# Effect of neoadjuvant chemotherapy on tumor immune infiltration in breast cancer patients: Systematic review and meta-analysis

**Manuela Llano-León**[1,2☉], **Laura Camila Martínez-Enriquez**[2☉], **Oscar Mauricio Rodríguez-Bohórquez**[2☉], **Esteban Alejandro Velandia-Vargas**[2☉], **Nicolás Lalinde-Ruíz**[2☉], **María Alejandra Villota-Álava**[2☉], **Ivon Johanna Rodríguez-Rodríguez**[2,3☉], **María del Pilar Montilla-Velásquez**[2☉], **Carlos Alberto Parra-López**[2☉] *

**1** Departamento de Química, Facultad de Ciencias, Universidad Nacional de Colombia, Bogotá, Colombia, **2** Laboratorio de Inmunología y Medicina Traslacional, Departamento de Microbiología, Facultad de Medicina, Universidad Nacional de Colombia, Bogotá, Colombia, **3** Departamento de Movimiento Corporal Humano, Facultad de Medicina, Universidad Nacional de Colombia, Bogotá, Colombia

☉ These authors contributed equally to this work.
* caparral@unal.edu.co

**Data Availability Statement:** All relevant data are within the paper and its Supporting Information

## Abstract

The tumor immune infiltrate has an impact on cancer control and progression, additionally a growing body of evidence has proposed the role of neoadjuvant chemotherapy in modulating the contexture of the tumor immune infiltrate. Here, we performed a systematic review to evaluate the effect of chemotherapy in the immune infiltration of breast cancer tumors. We systematically searched Pubmed/MEDLINE, EMBASE, CENTRAL, and BVS databases with a cutoff date of 11/06/2022. Studies in patients with pathological diagnosis of BC, whose first line of treatment was only NAC, were included. Only published experimental studies that measured tumor immune infiltrate before and after NAC by hematoxylin and eosin (H&E) staining, immunohistochemistry (IHQ), or transcriptome were included. Reviews, studies with animal models and *in-vitro* models were excluded. Studies in which BC was not the primary tumor or studies with patients who received other types of neoadjuvant therapy were also excluded. The NIH quality assessment tool for before and after studies without control was used. We included 32 articles that evaluated the proximal tumor microenvironment before and after neoadjuvant chemotherapy in 2072 patients who received NAC as first line of treatment and who were evaluated for immune infiltrate in the pre- and post-chemotherapy tumor sample. Results were divided into two major categories immune cells and *in-situ* expression of immune checkpoints and cytokines. Qualitative synthesis was performed with the 32 articles included, and in nine of them a quantitative analysis was achieved, resulting in six meta-analyses. Despite high heterogeneity among the articles regarding treatment received, type of tumor reported, and techniques used to evaluate immune infiltrate, we found a significant decrease of TILs and FoxP3 expression after neoadjuvant chemotherapy. The study protocol was registered in PROSPERO 2021 (Protocol ID: CRD42021243784) on 6/29/2021.

files. All information on how this systematic review was conducted is available in the methods section and the data extracted from each article is in supplementary material 3.

**Funding:** This work was funded through Dirección de Investigación de Bogotá (DIB)-HERMES Grants (Numbers 47334, 48357 and 50297) from the Universidad Nacional de Colombia and funds from the joint grant among Fundación Salud de Los Andes, Universidad Nacional de Colombia and MINCIENCIAS (No. 110177758253, No. 110184168973).

**Competing interests:** The authors declare that they have no conflicts of interest.

# Introduction

Tumor microenvironments are a group of interconnected layers organized at different levels, whose definition and delimitation are conceptually and experimentally challenging [1], however, for the purpose of this review, we defined tumor immune microenvironment (TIME) as the presence of leukocytes, as well as their products, surface markers, and gene expression profiles within or around the tumor. The presence, location, or functional organization of infiltrating immune populations exhibit high variability among tumors [2]. This heterogeneity suggests that different immune cell populations may have distinct roles in the control or promotion of tumor growth.

The study of these characteristics has allowed the identification of prognostic markers of the clinical response to treatment, leading to different approaches to classify neoplasms [2]. Classically, tumors have been staged using the TNM system, but recently, the validation of models such as *Immunoscore* has allowed a more refined evaluation of colorectal cancer patients prognosis, becoming a reference system for establishing clinical outcome by evaluating the density of CD3+CD8+ T lymphocyte infiltration within the tumor and at the invasive margin of the tumor [3].

In breast cancer (BC), the importance of immune infiltration as a prognostic factor for clinical response to neoadjuvant chemotherapy (NAC) has also been evaluated [4, 5]. Specifically, in these tumors it has been shown that in non-tumor cells the expression of certain genes involved with proliferation such as topoisomerase IIα, CDC2 and PCNA correlates with a low rate of metastasis [6]. Additionally, other studies evidenced that the expression of cytokine genes such as IL-12, IFNg, and IL-2 correlates with favorable prognosis, whereas the expression of cytokine genes such as IL-4, IL-13 and TGFb [7] or a higher number of Tregs are of poor prognosis [8].

Chemotherapy has traditionally been described as an immunosuppressive treatment; however, a growing body of evidence has shown that it can activate the immune system by inducing immunogenic cell death (ICD) in tumor cells, promoting the release of DAMPs [9–14]. According to Sistigu, et al., anthracycline-induced production of type I IFNs by tumor cells increases MHC-I expression in these cells [15] and stromal cells [16], contributing to tumor recognition, activation and maturation of antigen presenting cells (APCs), and ultimately enhancing T cells response to tumors. This phenomenon has also been described after treatment with chemotherapeutic agents such as Doxorubicin, Cyclophosphamide, Oxaliplatin or Gemcitabine [9, 11, 12, 17–20].

The aim of this review was to determine whether there are changes in the cellular and molecular component of the immune infiltrate in BC tumors in response to NAC. A systematic search of the literature was performed and studies comparing immune molecules or cells before and after treatment were included.

# Materials and methods

## Study design and protocol registration

This systematic review was based on the PRISMA criteria for reporting and item design for systematic reviews and meta-analyses. The study protocol was registered in PROSPERO 2021 (Protocol ID: CRD42021243784) on 6/29/2021.

## Search strategy

Studies indexed were systematically searched in the following databases: Pubmed/MEDLINE, EMBASE, CENTRAL, and BVS with a cutoff date of 11/06/2022. The search strategy included population (e.g., "Human", "Breast Neoplasm"), intervention ("Neoadjuvant chemotherapy"),

techniques (e.g., "Immunohistochemistry", "flow cytometry") and immune cells assessed (e.g., "T lymphocytes", "B lymphocytes", "neutrophil") with vocabulary and syntax adjusted across databases. Articles in English and Spanish were included. Advanced filters were used to eliminate reviews and animal studies (S1 Appendix).

## Eligibility criteria

In order to compare the immune infiltrate before and after NAC, studies in patients with pathological diagnosis of BC, whose first line of treatment was only NAC, were included. To reduce data variability and make accurate comparisons between articles, only published experimental studies that measured tumor immune infiltrate before and after NAC by eosin and hematoxylin (H&E) staining, immunohistochemistry (IHQ), or transcriptome were included. Reviews, studies with animal models and *in-vitro* models were excluded. Studies in which BC was not the primary tumor or studies with patients who received other types of neoadjuvant therapy (besides NAC, i.e., hormonal therapy, immunotherapy, etc.) were also excluded. In articles with mixed neoadjuvant therapy, data were extracted only from the cohort that did not receive it, or the triple negative BC cohort.

## Study selection and data collection process

Duplicate articles were eliminated, and two reviewers independently screened the titles and abstracts of the retrieved studies by applying the eligibility criteria. Subsequently, the articles included by title and abstract were read in full text to decide their inclusion in the review. In case of conflicts or disagreements about the eligibility of studies, the final decision was made by consensus. Finally, 32 articles met all the eligibility criteria. To facilitate the extraction, management, and storage of qualitative and quantitative data, excel tables were created. The bibliographic details of the selected articles are reported in S1 Table. Included articles.

The extracted data correspond to immune infiltrate in BC tumors before and after NAC. Clinical data of patients such as tumor type, treatment response and NAC schedule, were extracted when available. Measurement effects correspond to the difference between cell, immune checkpoint, or cytokine data before and after NAC.

Results were separated into the following domains: (i) cellular markers such as: tumor infiltrating lymphocytes (TILs), CD45, CD3, CD4, CD8, CD68, CD66b, CD163, CD20, FoxP3, CD56, CD83, CD1a, s100. (ii) molecular markers such as: PD1, PD-L1, CTLA4, LAG3, TIM3, IL-1, IL-2, IL-4, IL-10, IL-17, IFNg, TGFb and OX40. When available, clinical parameters regarding disease progression and pathologic response such as, but not restricted to: overall survival (OS), residual disease (RD), pathologic complete response (pCR), disease-free survival (DFS), major pathological response (MPR), breast cancer specific survival (BCSS) and good pathological response (GPR) associated with change in post-NAC infiltration were extracted.

For the management of unreported or missing data, the corresponding authors and the lead author were contacted electronically, requesting the raw data from the studies. In case of no response within two weeks, figures data was extracted using Graph Grabber v2.0.2.

## Risk of bias assessment

The risk of bias assessment was performed through the NIH quality assessment tool for before and after studies without control group, with the addition of three criteria: i) Explicit description of protocol evaluation and approval by an ethics committee, or if patients signed an informed consent, ii) Description of the antibodies, clones and commercial brands used, and iii) Explicit presentation of the results either in the text, tables, and graphs. This modified risk of bias table had 3 instances of independent review.

## Synthesis of results

Nine articles had the required data to perform meta-analysis. Information was synthesized into forest diagrams. A qualitative description of the remaining papers was performed and presented in narrative form using tables and figures.

## Statistical analysis

The effect measures analyzed corresponded to the mean counts of leukocyte populations, immune checkpoint (ICP), cytokines and cellular infiltrate together with their respective standard deviation. These parameters were used for the calculation of the standardized mean (SM) and standardized mean difference (SMD) of cell counts quantified from histopathological material pre- and post-NAC administration. Calculation of the effect estimator with 95% confidence interval was performed for each comparison. Weighting was performed using the inverse variance method. The effects were summarized in the random-effects meta-analysis. Effect size and confidence intervals were presented in forest plots. There was insufficient data for sensitivity and publication bias analysis. RevMan v.5.4 software was used for meta-analysis.

# Results

## Search results and study characteristics

The search strategy carried out led to an initial selection of 623 articles. In the identification phase, 257 articles were obtained from Pubmed, 76 from Medline, 150 from Embase, 75 from Cochrane, and 65 from BVS. After eliminating duplicates, 449 articles were identified. Of these studies, 317 were excluded based on title and abstract. Subsequently, open-text reading of the remaining 132 articles was performed which led to the inclusion of 32 studies for qualitative analysis (S1 Table), of which nine were used for meta-analyses (Fig 1. Prisma flowchart) and the exclusion of 100 articles (S2 Table).

The 32 studies included in this review were published between 2001 and 2022, including a total of 2072 patients older than 18 years, who received NAC as first line of treatment and who were evaluated for immune infiltrate in the pre- and post-chemotherapy tumor sample. The parameters measured in each article are described in S1 Table. The results were classified into two groups: cells (Fig 2A) and immune checkpoints or cytokines (Fig 2B). The cell populations or markers that were most frequently evaluated were TILs (53.1%), CD8 (48.6%), FoxP3 (48.6%), and PD-L1 (31.2%). Additionally, 19 of the 32 articles (59%) performed correlations between immune infiltrate changes and clinical response.

## Categorization of heterogeneity

The composition of tumor infiltrate status has been a widely studied topic in the pursuit of understanding which cell populations are desirable and prognostically good for patients [21]. For this reason, through this systematic review we sought to compare the largest number of studies that analyzed the immune infiltrate at baseline and after treatment with NAC to see how this altered the immune status of the tissue. In general, high heterogeneity was found in the methods used for measuring the different immune populations studied in the articles included. We found heterogeneity at two levels, the first corresponds to the technique: in the markers used to define the populations, and in the software for transcriptome analysis. And the second to the level of clinical information, where three elements stood out: i) The histopathological classification of BC, ii) the treatment received and iii) the spatial architecture of the microenvironment (S3 Table).

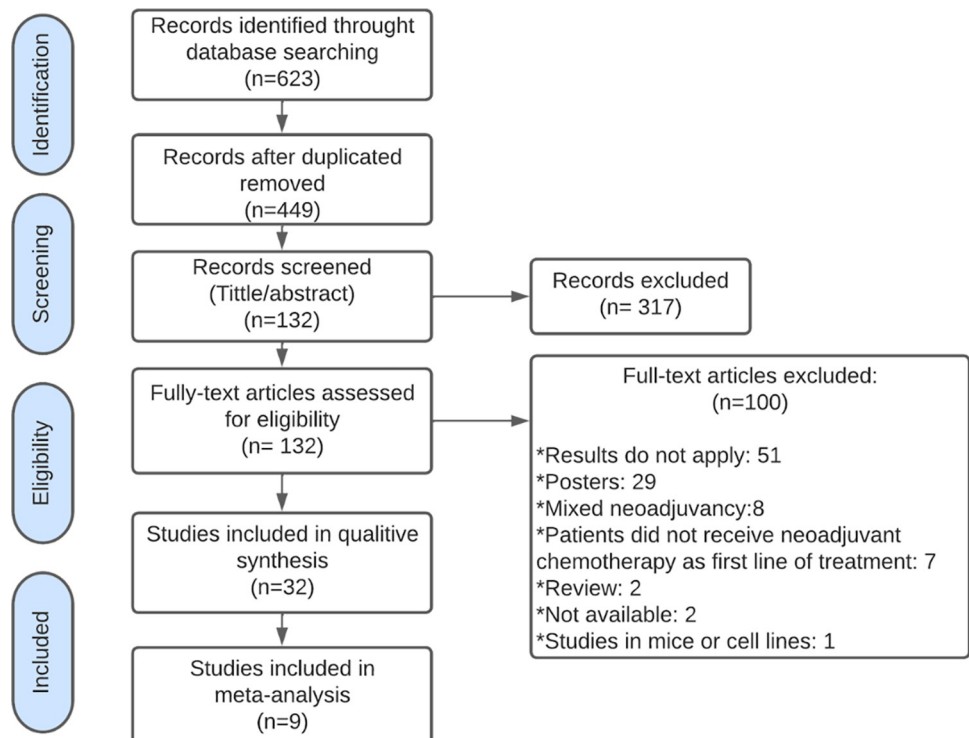

**Fig 1. Prisma study flow diagram.** The search strategy performed in this article led to the initial selection of 623 potential papers. After duplicates were removed, a total of 449 papers were identified. From these, 317 articles were excluded based on title and abstract. Further investigation of the remaining 132 full text articles led to the exclusion of 100 articles and the inclusion of 32 studies for the qualitative synthesis, with nine of them included in the meta-analysis.

## Risk of bias

Fig 3 shows that 6% of the studies had problems with the clarity of the study objectives. In 38% the selection and eligibility criteria for the population were not fully described; 31% had problems describing their selection criteria; and 34% had difficulty in describing the calculation of

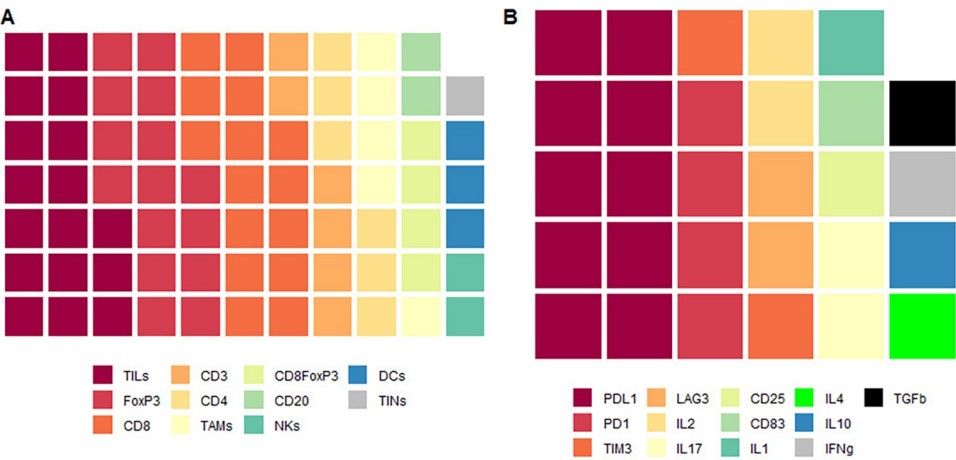

**Fig 2.** Waffle plot: Each box represents an article which measured a type of cell (A) or a marker (B).

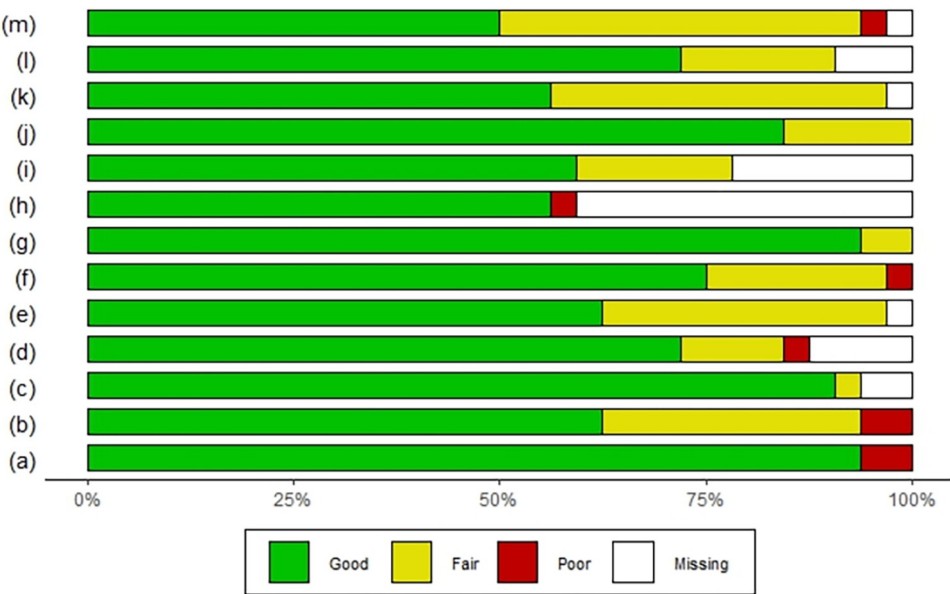

**Fig 3. Risk of bias summary.** Review authors' judgments about each risk of bias item for each included study. a) The study question and/or objective of the study is clear. b) Eligibility/selection criteria for the study population are specified and described in advance. c) Study participants were representative of the test/service/intervention in the general or clinical population of interest. d) Eligible participants who met the pre-specified entry criteria were enrolled. e) The sample size was sufficiently large to provide confidence in the findings. f) The test/service/intervention was clearly described and uniformly administered across the study population. g) Outcome measures were pre-specified, clearly defined, valid, reliable, and consistently assessed across all study participants. h) Individuals assessing outcomes were blinded to participants' exposures/interventions i) Loss to follow-up after baseline was 20% or less and loss to follow-up was accounted for in the analysis. j) Statistical methods for changes in outcomes from pre to post intervention were examined. Statistical tests were performed that provided p values for pre- to post-intervention changes. k) Whether an ethics committee approved the protocol, or whether patients signed an informed consent form is clearly described. l) The antibodies, clones and brand used in the labeling were described (For the technique that applies). m) Results were clearly presented.

the sample. 21% did not describe the interventions clearly; 40% did not clearly describe whether their protocol was reviewed and approved by an ethics committee, and 46% of the articles have difficulties in presenting the results.

## Tumor infiltrating lymphocytes

Tumor infiltrating lymphocytes (TILs) are lymphocytes that leave the bloodstream and migrate towards the tumor and can recognize and eradicate cancer cells [22]. Out of 32 articles included in this systematic review, 17 studies (53.1%) measured tumor infiltrating lymphocytes (TILs) by morphologic assessment of H&E slides [23–39]. Of these 17 articles, four reported a decrease in the percentage of TILs post-NAC [23–26], two reported an increase in TILs [27, 28] and 11 reported no significant changes between pre- and post-NAC samples (Table 1) [29–39]. Urueña, et. al., in addition to evaluating TILs morphologically, also evaluated the expression of CD45, a marker of total leukocytes by immunohistochemistry which did not change significantly before and after NAC [30] (Table 1).

Only two of the 17 articles were included in the TILs meta-analysis, which showed a significant decrease in post-NAC TILs (p<0.00001) with a mean difference of 0.85 (95% CI [0.50, 1.20]) and low heterogeneity (I2 = 0%) (Fig 4. TILs meta-analysis) [24, 26].

Of the articles that measured TILs pre- and post-NAC, only six made correlations between the change in TILs and patients' clinical response to treatment [25, 27–29, 35, 36]. Dieci, et. al.,

**Table 1. Distribution of the number of articles (n) according to the trend (low, no difference, high) for each of the analyzed cell.**

| Cell | n | % | Lower | Cell | n | % | No difference | Cell | n | % | Higher |
|---|---|---|---|---|---|---|---|---|---|---|---|
| FoxP3 | 9 | 69,23 | | TILs | 10 | 76,92 | | CD8 | 6 | 46,15 | |
| TILs | 2 | 15,38 | | CD3 | 5 | 38,46 | | CD4 stromal | 3 | 23,08 | |
| CD8 | 2 | 15,38 | | CD8 | 5 | 38,46 | | T cells | 2 | 15,38 | |
| CD4/CD8 | 1 | 7,69 | | FoxP3 | 4 | 30,77 | | CD4 Tumoral | 2 | 15,38 | |
| CD4 | 1 | 7,69 | | CD4/CD8 | 3 | 23,08 | | TILs | 1 | 7,69 | |
| M2 | 1 | 7,69 | | CD4 | 2 | 15,38 | | CD4 | 1 | 7,69 | |
| M1 | 1 | 7,69 | | CD68 | 2 | 15,38 | | FoxP3 Stromal | 1 | 7,69 | |
| TILLs responders | 1 | 7,69 | | CD8/FoxP3 | 2 | 15,38 | | FoxP3 tumoral | 1 | 7,69 | |
| FoxP3 non-responders | 1 | 7,69 | | M2 | 2 | 15,38 | | CD3 | 1 | 7,69 | |
| FoxP3 responders | 1 | 7,69 | | TILs non-responders | 2 | 15,38 | | CD8/FoxP3 | 1 | 7,69 | |
| LB | 1 | 7,69 | | CD1a Intraepithelial DCs | 1 | 7,69 | | CD1a Stromal DCs | 1 | 7,69 | |
| M2 Stromal | 1 | 7,69 | | CD1a Stromal DCs | 1 | 7,69 | | CD8 Stromal | 1 | 7,69 | |
| FoxP3 Stromal | 1 | 7,69 | | CD20 | 1 | 7,69 | | TILs Stromal | 1 | 7,69 | |
| TMB | 1 | 7,69 | | CD3, CD8, FoxP3 | 1 | 7,69 | | TILLs Tumoral | 1 | 7,69 | |
| FoxP3 tumoral | 1 | 7,69 | | CD45 | 1 | 7,69 | | CD8 tumoral | 1 | 7,69 | |
| | | | | CD8 Intratumoral | 1 | 7,69 | | CD3 Intraepithelial | 1 | 7,69 | |
| | | | | CD8 Stromal | 1 | 7,69 | | CD3 stromal | 1 | 7,69 | |
| | | | | CD83 Intraepithelial DCs | 1 | 7,69 | | CD8 non-responders | 1 | 7,69 | |
| | | | | Immune index | 1 | 7,69 | | CD8 responders | 1 | 7,69 | |
| | | | | M2 Intraepithelial | 1 | 7,69 | | CD8/FoxP3 non-responders | 1 | 7,69 | |
| | | | | NK Intraepithelial | 1 | 7,69 | | CD8/FoxP3 responders | 1 | 7,69 | |
| | | | | TINs Intratumoral | 1 | 7,69 | | CD83 Stromal DCs | 1 | 7,69 | |
| | | | | M1 | 1 | 7,69 | | DC | 1 | 7,69 | |
| | | | | NK | 1 | 7,69 | | DCs 100 Stromal | 1 | 7,69 | |
| | | | | Stromal TINs | 1 | 7,69 | | CD1a Intratumoral DCs | 1 | 7,69 | |
| | | | | TILs responders | 1 | 7,69 | | NK Stromal | 1 | 7,69 | |
| | | | | TILs Stromal | 1 | 7,69 | | Stromal T cells | 1 | 7,69 | |
| | | | | TILs Tumoral | 1 | 7,69 | | T cells tumoral | 1 | 7,69 | |
| | | | | CD8 tumoral | 1 | 7,69 | | | | | |
| | | | | CD4+/CD8+ Tumoral | 1 | 7,69 | | | | | |

reported correlations between increased TILs in post-NAC samples and higher DFS in RD patients [35]. In contrast, Li reported that, in patients with pCR the TILs count was significantly lower in post-NAC samples, while in RD patients there was no significant difference between pre- and post-NAC TILs [25]. Additionally, the remaining four articles found no

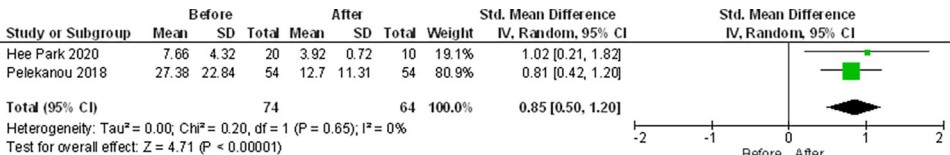

**Fig 4. TILs meta-analysis.** Influence of NAC in the TILs density of BC tumors. Forest plot for the different outcomes regarding TILs density before and after NAC. The forest plot displays the SMD (squares) and 95% confidence interval of the individual studies. The diamond in each plot indicates the overall estimate and 95% confidence interval.

significant associations between change in TILs pre- vs post-NAC and clinicopathological variables [27–29, 36]

## Macrophages

Macrophages (Mφ) are a highly heterogeneous population and their characterization using membrane proteins is not fully defined yet. Canonically, there are two types of Mφ: M1 Mφ, which have a pro-inflammatory profile and are considered desirable because of their potent antitumor activity [40]. On the other hand, M2 Mφ are anti-inflammatory and promote angiogenesis, which favors tumor growth [41]. Of the 32 articles included in the review, 5 (15%) measured the infiltration of these cells [29, 30, 42–44]; however, the definition of this cell population, according to the markers reported, varied between studies. Four of these articles used CD68; in particular, Urueña, et al., Garcia-Martinez, et al., and Hornychova, et al used this marker to define total Mφ [30, 42, 43], while Kaewkangsadan, et al., [44] used CD68 to delimit M1-Mφ (pro-inflammatory), and CD163 for M2-Mφ (anti-inflammatory) [44]. Waks, et. al., [29], measured this population with the NanoString method, using the PanCancer IO360 panel, but due to the high heterogeneity of Mφ, one set of genes associated with an M1 response and another set for an M2 response were determined (Table 1).

Globally, there was no consensus among the articles that measured the infiltration of these cells. Four of the five articles used IHC for the determination of infiltrate; in two of them a decrease of tumor infiltrating Mφ in response to NAC was reported [42], although in one of them this decrease was observed at the stromal and not at the intraepithelial level [43], in the remaining two articles no significant differences were observed [30, 44]. In the study that used NanoString, an overall increase in Mφ was found, but when looking specifically at genes associated with a pro-inflammatory response, cytokine genes such as CXCL9 and TNF, and transcription factor Notch1 were decreased post-NAC, while receptor genes such as TLR4, CD40, Fas and IL-15 were increased [29].

Regarding clinical response, only two articles correlated pCR and OS of patients with the change of pro-inflammatory Mφ immune infiltrate and in neither case found statistical significance [42, 44].

On the other hand, anti-inflammatory Mφ infiltration was measured in two of the four articles [29, 44]; in one of them no significant change in response to NAC was found [44]. In contrast, Waks et al. observed an increase in CD163 in post-NAC samples, accompanied by an increase in chemokine and cytokine genes related to this phenotype such as IL-10, TGFb, CCL2, CCL13, CCL14; membrane markers such as CD36, CD209, CD115, MRC1, and transcription factors such as STAT3 and PPARG [29].

Only Kaewkangsadan's work correlated M2-Mφ infiltrate with clinical response, and it was found that high levels of TIMs (CD163+) in BC were significantly associated with a GPR and pCR after 8 cycles of NAC [44]. Likewise, a significant association could also be found with ER status and tumor grade, which were positively associated with pCRs.

## Neutrophils

Neutrophils have been recognized as the most abundant innate immune cell in the bone marrow and peripheral blood. In cancer patients, blood neutrophils have been associated with increased free radical production and poor clinical prognosis [45]. Only 1 article (3.1%) measured the presence of Tumor Infiltrating Neutrophils (TINs), using the CD66b marker, both intratumoral and stromal, in two groups of patients: with and without metastasis [44]. This analysis reported no significant difference in the infiltration of this cellular subtype between pre- and post-NAC samples in either of them (Table 1). Additionally, no correlation was found between CD66b+ cell change and pathologic response.

## Natural Killer cells

Natural Killer (NKs) cells are characterized by containing cytoplasmic granules that can destroy cancer cells and impact the adaptive anti-tumor immune response by producing chemokines and cytokines. Additionally, NK cells can have a negative impact in anti-cancer responses by modulating DC and T cells [46]. NKs cells were measured by IHC through the CD56 marker in two of the 32 articles (6.2%) [39, 43]. Verma, et. al., reported no change in response to NAC [39]. On the other hand, Hornychova found a significant increase in NKs cells at the stromal level, but not at the intraepithelial level (Table 1) [43]. Neither found a significant association between NK infiltration change in response to NAC and clinical response.

## Dendritic cells

Dendritic cells (DCs) are part of the innate immune system responsible for initiating the activation of T cells through antigen presentation (86). However, DCs are highly heterogeneous cells and depending on their expression of surface markers can activate an adequate antitumor response or, on the contrary, induce a state of anergy and tolerance that favors tumor growth [47–49]. DCs were evaluated in three of the 32 articles (9.3%) [29, 43, 44]. Within the review we could find six different markers: CD1a, CD83, S100, CD209, CCL13, HSD11B to identify DCs. The CD1a marker, used to describe immature DCs, was evaluated intratumorally and stromally in two articles, finding in the case of Kaewkangsadan that there was a significant post-NAC decrease at intratumoral level [44], while Hornychova reported a significant increase at stromal level [43]. Using CD83, as a maturation marker, no change in the infiltration of mature DCs was reported at the intraepithelial level, but a significant post-NAC increase was reported at the stromal level. S100 protein was evaluated in Hornychova reporting a significant increase only at the stromal level [43]. On the other hand, in the Waks study the evaluation of DCs by NanoString was performed, which defines this population by CD209, CCL13 and HSD11B markers, finding that there was a significant increase in DCs post-NAC [29]. Taking these results together, it is possible to say that there is an increased presence of DCs in the tumor after NAC (Table 1). No analysis was made between the change of this population and the clinical response of the patients.

## B cells

B cells are responsible for the production of antibodies that favor an anti-tumor response by promoting tumor killing by NK cells, phagocytosis by macrophages, and the priming of CD4 + and CD8+ T cells. However, B regulatory cells subpopulation has been related with the development of metastasis of breast cancer cells [50]. In only two articles of the 32 analyzed B cells infiltration was measured (6.2%) [30, 42]. For this, Garcia-Martinez and collaborators used the CD20 marker to define this cell population and reported a significant decrease in response to NAC [42]. In contrast, Urueña, et. al., using the same marker observed no change in response to NAC (Table 1) [30]. For the decrease reported by Garcia-Martinez, a significant association with improved pCR was found [42]. However, no correlation was found between the decrease of these lymphocytes with the OS of the patients.

## T cells

T lymphocytes are the most abundant type of cells of the adaptive immune system. These cells are classified in two main subpopulations, CD4 helper and CD8 cytotoxic T cells. Their response depends on the presentation of antigens by DCs, Mφ and B lymphocytes. The cytotoxic CD8+ T cells are assisted by CD4+ T helper cells through the secretions of cytokines, that build up the anti-tumor response [22]. The impact of the infiltration of these cells is

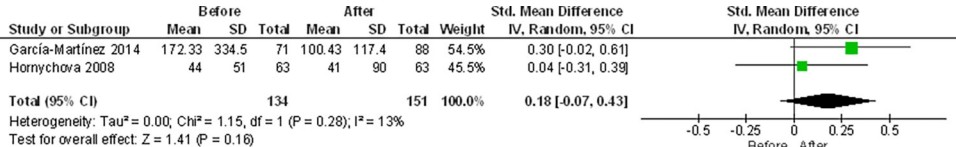

**Fig 5. CD3+ T cells meta-analysis.** Influence of NAC in the CD3+ T cells infiltration of BC tumors. Forest plot for the different outcomes regarding CD3+ T cells infiltration before and after NAC. The forest plot displays the SMD (squares) and 95% confidence interval of the individual studies. The diamond in each plot indicates the overall estimate and 95% confidence interval.

controversial since in some models such as oropharyngeal cancer is desirable, while in colon, ovarian and cervical cancer represents poor prognosis [51–57]. The determination of T cells was done using the CD3 marker and six of the 32 articles included (18.7%) measured the infiltration of these cells [30, 34, 42, 43, 58, 59]. Two of these papers reported increased infiltration of T cells in post-NAC samples [43, 60]. In the other four articles no significant differences were found between pre and post-NAC samples (Table 1) [34, 42, 44, 58], although Demaria, et. al., described that there was a change in T cell localization, with pre samples being in the stroma and post-samples intraepithelial [34].

Likewise, two of the six articles were included in the meta-analysis (Fig 5. T cell meta-analysis) [42, 43]. No significance (p = 0.16) was found in post-treatment infiltration compared to infiltrate in pre-NAC samples, with a mean difference of 0.18 (95% CI [-0.07, 0.43]) and low heterogeneity (I2 = 13%).

Regarding the relationship between change in T cell infiltration and clinical response, only in two papers these analyses were performed [42, 44], and only Garcia-Martinez et al. found an association in the decrease of these cells in response to NAC with improved pCR and improved survival measured in OS and DFS [42].

## CD4+ T cells

CD4+ T helper cells coordinate the immune response by triggering CD8 T cells, B cells and Mφ through the secretions of different cytokines such as TNF-a, IFNg, IL, 6, IL-10, IL-17. This population can be differentiated into several subpopulations depending on the condition and stimuli [61]. Measurement of CD4+ helper T cells was performed in six articles (18.75%) [28, 30, 37, 42, 60, 62], of which two reported a decrease in response to NAC [37, 42], two did not observe significant changes overall [30, 60]. Although one reported a significant decrease in patients who showed a GPR [60]. Finally, Lee, et. al., found a significant increase in this population after NAC (Table 1) [28].

For the meta-analysis performed in this review, only the works of Garcia-Martinez, et. al., and Lee, et. al., could be included, and we did not find a statistically significant difference (p = 0.09), as these both had contrasting findings (Fig 6. CD4+ T cell meta-analysis) [28, 42]. The mean difference found was 0.3 (95% CI [-0.05,0.65], with relatively low heterogeneity (I2 = 36%). Thus, no pattern of behavior could be established for the infiltration of these cells in response to NAC.

Again, only Garcia Martinez evaluated the correlation between the decrease of these cells and clinical response, finding that patients who had a lower presence of CD4+ T cells post NAC had a better pCR [42].

## CD8+ T cells

CD8+ T cytotoxic cells play a central role in anti-tumor response through their capacity to eliminate cancer cells upon recognition of tumor antigens. These are one of the main cells that

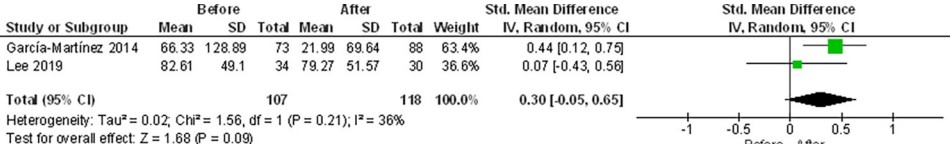

**Fig 6. CD4+ T cells meta-analysis.** Influence of NAC in the CD4+ T cells infiltration of BC tumors. Forest plot for the different outcomes regarding CD4+ T cells infiltration before and after NAC. The forest plot displays the SMD (squares) and 95% confidence interval of the individual studies. The diamond in each plot indicates the overall estimate and 95% confidence interval.

infiltrate the TIME in order to control tumor growth [63]. CD8+ cytotoxic T cells were evaluated in 15 of 32 articles (48.6%) [23, 28–30, 35, 37, 42, 58–60, 64–68]. Three reported a decrease in this population after NAC [23, 59, 64], seven reported an increase after treatment [28, 35, 42, 60, 66–68], and the remaining five found that there was no change after NAC [29, 30, 37, 58, 65]. Half of the studies that evaluated this marker did not report a p-value of the tests performed (Table 1) [23, 28, 29, 58, 65, 67].

Regarding this population, it was possible to perform several meta-analyses considering the distribution in the TIME. First comparison was made in three articles of the total CD8+ count, observing that there was no significant change of these cells post-NAC, with a mean difference of -0.17 (95% CI [-0.46, 0.13], p = 0.27) and low heterogeneity (I2 = 0.27%) [23, 28, 42]. Subsequently, an analysis was performed for the tumor infiltrating CD8+ fraction in two articles, resulting in no significant change in CD8+ infiltration post-NAC, with a mean difference of 0.40 (95% CI [-0.16, 0.96], p = 0.16) and low heterogeneity (I2 = 0%) [23, 37]. Additionally, with two articles, a comparison was made for the stromal CD8+ fraction, again obtaining no change in the count of these cells pre and post-NAC with a mean difference of 0.42 (95% CI [-0.15, 0.96], p = 0.18) and low heterogeneity (I2 = 0%)) [23, 37]. Finally, an analysis was made considering all the articles without differentiating the location, where it was corroborated that there is no change in these cells in response to NAC, with a mean difference of 0.06 (95% CI [0.22, 0.33], p = 0.69) and a low heterogeneity (I2 = 27%) (Fig 7. CD8+ T cell meta-analysis-count).

Ladoire and Wang grouped patients according to the degree of infiltration prior to NAC [65, 68]. The meta-analysis found no significant change in the degree of infiltration in response to treatment among patients with high pre-NAC infiltration (mean difference of -0.03 (95% CI [-0.19, 0.14], p = 0.76) and low heterogeneity (I2 = 56%)). The same was seen in patients with low pre-NAC infiltration (mean difference of 0.02 (95% CI [-0.16, 0.20], p = 0.82) and low heterogeneity (I2 = 60%)). Finally, pooled analysis of these groups showed no significant change in infiltration in response to NAC (mean difference of -0.00 (95% CI [-0.11, 0.11], p = 0.97) and low heterogeneity (I2 = 52%) (Fig 8. CD8+ T cell meta-analysis-count).

Six articles associated CD8+ T cell infiltration with clinical outcome [42, 59, 60, 65–67]. Garcia-Martinez and Chan did not find correlation between pCR and CD8+ [42, 66], while Vanguri showed a significant decrease in CD8+ T cell infiltration only in patients with pCR [59], furthermore Liang reported an increase only in non-responders [60]. Additionally, Miyashita and Ladoire 2011 found a positive correlation between a high rate of change in CD8 + and relapse free survival (RFS) [65, 67].

## FoxP3

FoxP3 regulatory T cells are an immunosuppressive subpopulation of CD4+ T cells that modulate the anti-tumor response mediated by CD8+ T cells. In the TIME, high levels of Tregs are frequently presented and their main effect is to create and maintain an immune tolerant

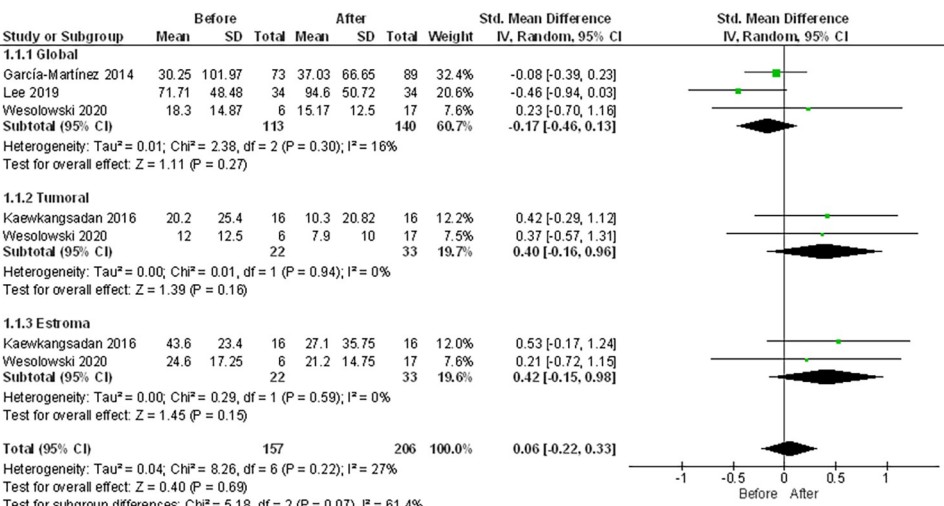

**Fig 7. CD8+ T cells count meta-analysis.** Influence of NAC in the CD8+ T cells count of BC tumors. Forest plot for the different outcomes regarding CD8+ T cells count before and after NAC. The forest plot displays the SMD (squares) and 95% confidence interval of the individual studies. The diamond in each plot indicates the overall estimate and 95% confidence interval.

environment [69]. 15 studies measured Tregs using the FoxP3 marker (46.8%) [28–31, 35, 37, 42, 58, 59, 64–67, 70, 71]. The majority (73%) showed a decrease after NAC [28, 29, 37, 58, 59, 64–66, 70, 71], while only 17% reported no difference from pre-NAC values (Table 1) [30, 31, 35, 42, 67]. Meta-analysis could be performed in five articles, finding a decrease in the expression of this marker post-NAC (mean difference of 0.45 (95% CI 0.05–0.86 P = 0.03) and high heterogeneity (I2 = 76%)) (Fig 9. Foxp3+ cells meta-analysis-count) [28, 37, 42, 66, 70].

Five studies associated changes in FOXP3 post-NAC with clinical response [42, 65–67, 70]. In three articles, no relationship was found between FOXP3 infiltrate changes and clinical outcomes [42, 66, 67]. Garcia-Martinez, et. al., evaluated association with pCR [42], Chan with DFS and OR [66], and Miyashita with RFS and BCSS [67]. In contrast, Ladoire evidenced that decreased FOXP3 post-NAC is associated with better RFS [65] and Demir found that patients with lower FOXP3 expression had a higher pCR rate [70].

## CD8/FoxP3 ratio

Four articles evaluated CD8/FoxP3 ratio (12.5%) [28, 65–67]. Two studies reported an increase [28, 66], while the other two reported no change [65, 67]. It is important to highlight that since

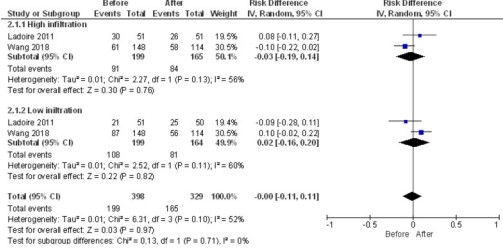

**Fig 8. CD8+ T cells infiltration meta-analysis.** Influence of NAC in the CD8+ T cells infiltration of BC tumors. Forest plot for the different outcomes regarding CD8+ T cells infiltration before and after NAC. The forest plot displays the SMD (squares) and 95% confidence interval of the individual studies. The diamond in each plot indicates the overall estimate and 95% confidence interval.

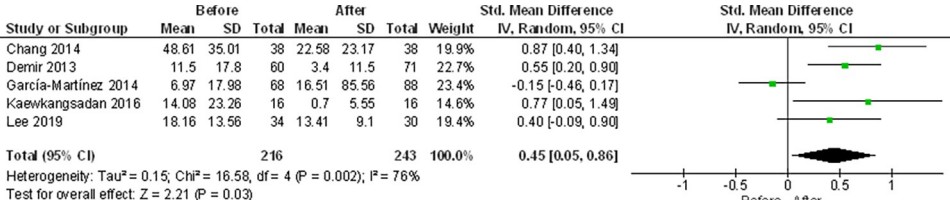

**Fig 9. FoxP3+ Tregs infiltration meta-analysis.** Influence of NAC in the FoxP3+ Tregs infiltration of BC tumors. Forest plot for the different outcomes regarding FoxP3+ Tregs infiltration before and after NAC. The forest plot displays the SMD (squares) and 95% confidence interval of the individual studies. The diamond in each plot indicates the overall estimate and 95% confidence interval.

only one of the articles reported p-value, it was not possible to infer statistical significance (Table 1) [66].

Three of the aforementioned studies made associations between clinical outcomes and changes in CD8/FoxP3 ratio [65–67]. Chan, et. al., reported data categorizing according to the pathological response evidenced, dividing the population into a responder group and a non-responder group [66]. In both groups a significant increase in the CD8/FoxP3 ratio post-NAC was found, with change being greater in the responder group. On the other hand, Ladoire, et. al., did not show changes in CD8/FoxP3 ratio post-NAC in patients with HER2- BC [65]. However, when following up for five years to evaluate RFS and OS, they identified that patients with an increase in the CD8/FoxP3 ratio post-NAC had a significant improvement in these clinical outcomes compared to patients who did not, regardless of their pathological response. Miyashita, et. al., reported data as the percentage of patients with an increased ratio [67]. When performing an evaluation of the relationship of CD8/FoxP3 ratio with RFS and BCSS, it is reported that a greater change in CD8/FoxP3 ratio after NAC is significantly associated with better RFS and BCSS.

### Immune checkpoints and cytokines

Immune checkpoint molecules and cytokines have been identified as crucial regulators of the immune response. However, the prognostic significance of these molecules remains controversial. Establishment of an immunosuppressive environment within the tumor is regulated by diverse immune checkpoint molecules and cytokines triggered by cancer cells or regulatory immune cells [72].

Among the immune checkpoints included in this review were PD1 and its ligand PD-L1, CTLA4, TIM3 and LAG3. PD1 and CTLA4 have been the most well-studied markers within the field and both are expressed on T cells [73]. The former interacts with its ligands PD-L1 and PD-L2 and induces apoptosis in exhausted T cells [74], while the latter interacts with the B7 molecules expressed on dendritic cells, preventing an adequate activation of T cells [75]. On the other hand, TIM3 and LAG3 are novel markers that are under extensive research for their association with a senescent phenotype [76] and hence, could also have a potential role in immune escape of cancer [77].

13 studies included (40.6%) evaluated immune checkpoints or cytokines in the tumor infiltrate (Table 2) [23, 24, 28, 31, 36–39, 60, 62, 64, 68, 71]. The CTLA4 marker was only evaluated by Kaewkangsadan, et. al., (3.1%) and they reported a significant reduction of its expression in post-NAC stromal T cells with no changes in intratumoral T cells [37].

PD1 was evaluated in four articles (12.5%) [23, 37, 60, 68]. Wang, et. al., reported a significant increase in PD1 expression in stromal CD8+ T cells [68]. Kaewkangsadan, et. al., showed a reduction in both intratumoral and stromal PD1+ T cells [37], while Wesolowski, et. al.,

**Table 2. Distribution of the number of articles (n) according to the trend (low, no difference, high) for each of the analyzed immune checkpoint.**

| Checkpoints | n | % | Lower | Checkpoints | n | % | No difference | Checkpoints | n | % | Higher |
|---|---|---|---|---|---|---|---|---|---|---|---|
| CTLA-4 stromal | 1 | 7,69 | | PD-L1 | 7 | 53,85 | | PD-L1 tumoral | 1 | 7,69 | |
| PD1 stroma | 1 | 7,69 | | PD-L1 tumoral | 3 | 23,08 | | PD1 | 1 | 7,69 | |
| PD1 tumoral | 1 | 7,69 | | PD-L1 in stroma | 2 | 15,38 | | TIM3 | 1 | 7,69 | |
| PD-L1 in stroma | 1 | 7,69 | | PD1 | 2 | 15,38 | | PD1 stroma in CD8 | 1 | 7,69 | |
| | | | | PD1 stroma in CD4 | 1 | 7,69 | | PD1 in CD8 | 1 | 7,69 | |
| | | | | PD1 tumoral in CD4 | 1 | 7,69 | | PD-1 CD8 | 1 | 7,69 | |
| | | | | CTLA-4 tumoral | 1 | 7,69 | | LAG3 | 1 | 7,69 | |
| | | | | PD-L1 TILS | 1 | 7,69 | | | | | |
| | | | | TIM3 | 1 | 7,69 | | | | | |
| | | | | PD-1 CD4 | 1 | 7,69 | | | | | |
| | | | | TIM3 CD8 | 1 | 7,69 | | | | | |
| | | | | TIM3 CD4 | 1 | 7,69 | | | | | |

reported a reduction in the number of patients whose tumors were positive for total PD1 (tumor and stromal) post-NAC [23]. Liang, et. al., described a significant reduction in PD1 in patients achieving pCR but this change was not evidenced in total patients [60]. When comparing PD1 expression in T cell subpopulations, Liang, et. al., reported a decrease in its expression in CD4+ T cells in the responder cohort. In contrast, PD1 expression in CD8+ T cells increased both globally and in the non-responder cohort [60].

PD-L1 was evaluated in 10 articles (31.25%) [23, 24, 28, 31, 36–38, 62, 68, 71]. Lee reported a significant increase in stromal cells but no change in tumor cells [28]. Wesolowski, et. al., found a reduction in the number of patients whose tumors were positive for PD-L1 at the total, tumor, and stromal levels [23]. On the contrary, other studies showed no significant differences in PD-L1 expression after NAC [24, 31, 36–38, 68, 71]. Sarradin, et. al., and Grecco-Hoffman, et. al., evaluated stromal PD-L1 expression [36, 38] while Wang, et. al., did it for stromal CD8+ T cells [68]. Interestingly, Zhang, et. al., used a combined positivity score, which evaluates the number of PD-L1-positive tumor cells, lymphocytes and Mφ over total tumor cells, finding a reduction of PD-L1 after NAC which was not found when compared directly (Table 2) [71].

LAG3 was evaluated in two studies (6.2%) [38, 68]. Sarradin, et. al., reported a significant decrease in stromal LAG3 expression after NAC [38]. In contrast, Wang, et. al., evaluated LAG3 expression in CD8+ T cells and reported an increase in its expression in patients who did not have a favorable response to NAC [68].

Two articles (6.2%) evaluated TIM3 [38, 60]. Sarradin, et. al., reported a significant stromal increase after NAC [38], while Liang, et. al., reported significant decrease only for non-MPR patients but not globally; nor when evaluating its expression segregating CD4+ and CD8+ T cells, or by clinical response in these subgroups [60].

Four articles associated PD-L1 change after NAC to clinical response [24, 28, 29, 68]. Wang, et. al., reported a negative association between PD-L1 increase after NAC and pCR [68]. In contrast, Waks, et. al., observed an association between good prognosis and an increase in tumor and stromal PD-L1 [29]. In addition, Lee, et. al., found that the absence of PD-L1 or the loss of PD-L1 expression in response to NAC is associated with worse prognosis [28]. Pelekanou, et. al., did not find significant associations between change of PD-L1 expression in response to NAC and clinicopathological variables [24]. Only Wang, et. al., reported a

**Table 3. Distribution of the number of articles (n) according to the trend (low, no difference, high) for each of the analyzed cytokines.**

| Cytokines | n | % | Lower | Cytokines | n | % | No difference |
|-----------|---|---|-------|-----------|---|---|---------------|
| IL-4 | 1 | 7,69 | ■ | IL-2 | 2 | 15,38 | ■ |
| | | | | IFNg | 2 | 15,38 | ■ |
| | | | | TGFb | 2 | 15,38 | ■ |
| | | | | IL-1 | 1 | 7,69 | ■ |
| | | | | IL-10 | 1 | 7,69 | ■ |
| | | | | IL-17 | 1 | 7,69 | ■ |
| | | | | IL-17F | 1 | 7,69 | ■ |

negative correlation between LAG3 expression and pCR, additionally when PD-L1 and LAG3 expression increased together patients exhibited worse prognosis [68].

Another key aspect of the immune response to cancer are the cytokines produced and secreted by the different cells, both immune and non-immune, in the tumor site. IL-2, IFNg, and IL-1 have canonically been associated with a Th1 response and good prognosis [78, 79], although this can vary depending on the tumor type [80]. On the other hand, cytokines such as IL-4, TGFb, IL-10 and IL-17 are mainly produced by Th2/Treg response and are thought to promote cancer progression [78, 79].

Kaewkangsadan, et. al., Verma, et. al., and Naofumi Oda, et. al., (9.3%) evaluated the change of cytokine expression *in-situ* in response to NAC [37, 39, 64]. Only IL-4 significantly decreased after NAC [37]. None of the three articles found significant changes in the expression of IL-2, IFNg, IL-1, TGFb, IL-10 or IL-17 after NAC (Table 3). Verma, et. al., reported no significant change in TGFb expression in patients whose tumors did not have pCR [39] (48). Lastly, Naofumi Oda, et. al., measured IL-17F and did not find an association between pCR and its expression after NAC [64].

### Change in TIME after one cycle of NAC

The change in cellular or molecular markers before and after a single cycle of NAC was measured in four articles [26, 28, 62, 81]. Two of them reported a significant increase in TILs between pre- and post-NAC samples [26, 28]. Lee, et. al., found no associations with significant clinicopathologic variables [28] and Hee-Park, et. al., did not perform this correlation [26]. The other two articles evaluated CD4+ and CD8+ T cell populations in both stroma and tumor tissue and reported a significant increase after the first cycle of NAC in both populations [28, 62]. In addition, Lee, et. al., quantified the population of Tregs by FoxP3 expression finding a significant decrease post NAC; furthermore, when evaluating the CD8/FoxP3 ratio they reported an increase after treatment [28]. Graeser, et. al., assessed total T cells, PD1 on CD4+ and CD8+ T cells; and PD-L1 on stromal and tumor cells [62]. Total T cell population increased after the first cycle of NAC. Regarding PD1 on CD4+ and CD8+ T cells, the former did not show a significant difference, while the latter exhibited a significant increase in its expression. Concerning PD-L1, a significant increase in its expression in tumor tissue was reported; however, no significant change was evidenced in the stroma. Lastly, Varadan, et. al., measured the "Immune index" which is defined as an immunological signature according to expression of selected genes, finding no significant change after the first cycle of NAC [81].

In summary, out of 623 articles originally found with our search strategy, 32 were included in this review and only nine could be used to perform meta-analyses (Fig 1). Our results showed a significant decrease in TILs post-NAC (Fig 4). The most studied cells in the articles

included were macrophages and T cells. Other cell types such as neutrophils, dendritic cells or B cells remain mostly excluded within the field of immune infiltration in cancer.

For T cells, meta-analyses showed no significant difference for neither CD4 (Fig 6) nor CD8 (Figs 7 and 8) T cells, being these cells the more frequent cellular subtype to be measured in immune infiltration. The last meta-analysis performed was for Treg cells, which showed a reduction of these cells after NAC (Fig 9). Unfortunately, no meta-analysis could be performed for inhibitory checkpoints or cytokines due to the high heterogeneity found in the articles included, which highlight the importance of using a broader spectrum of biomarkers as well as a need for technique standardization while studying immune infiltration in cancer.

## Discussion

The study of the TIME provides clarity on the physiology of cellular and molecular interactions that determine to some extent the establishment and progression of cancer, while offering clinical opportunities at the diagnostic, prognostic, and therapeutic levels [1].

McAllister and Weinberg, et. al., in 2014 proposed the concept of a tumor-induced systemic environment which determines cancer progression and metastatic capacity [82]. Laplane, et. al., in 2018 further develops this idea and posits the existence of six layers of the TME: the first comprising interactions and symbiosis between tumor cells. The second, the niche, which they define as a spatial/temporal concept that allows the initiation of tumor development. The next three layers are the confined, proximal, and peripheral environments, which while distant from the tumor center, also define the progress or regression of cancer [1]. Finally, they propose the tumor-organism environment (TOE) as the last layer, including components not located intratumorally, nor in its vicinity, which affect cancer development [1].

These definitions entail conceptual challenges, since it would be desirable to theorize about the communications between different organizational levels and, at the same time, to uncover new therapeutic targets. Because tumor progression depends on the establishment of all these layers, blocking or restricting any of them could stop tumor growth [1]. For example, Nalio, et. al., showed in BC that the presence of tissue resident FOLR2+ Mφ was correlated with good prognosis, meanwhile TREM+ Mφ, derived from infiltrating monocytes that differentiate into Mφ in the TIME were associated with poor prognosis [83].

Some authors have termed this new scenario, which includes immune structures distant from the tumor (such as lymph nodes and bone marrow), the nervous system, and the role of the microbiome in cancer development as the tumor-organism environment (TOE) [84]. Although the study of the TOE, and not only the tumor microenvironment will offer in the future more productive diagnostic and therapeutic tools in several types of cancer, the study of the TIME and particularly the proximal immune infiltrate in BC has proven to be cost-effective in the context of NAC, so an integrative analysis such as the one we propose here is relevant given the high heterogeneity evidenced.

The heterogeneity is partly explained by the current classification of BC that depends mainly on the expression of hormone and growth factor receptors, as well as on the macroscopic spatial architecture of tumors [85, 86]. This implies that several features useful for classification, such as the level of TILs or transcriptome analyses, which have prognostic relevance, are neglected [4, 5]. The integration of these features has allowed the development of new transcriptome analysis platforms, such as TNBC subtyping which have refined the classification of triple negative BC [87].

NAC is the first line of treatment for BC, which has been shown to have a significant effect in reducing tumor size prior to surgery. Classically, it has been described as an immunosuppressive treatment because of its cytotoxic effect on circulating immune populations, however,

a growing body of evidence has shown that it can activate the immune system, inducing ICD in tumor cells [88, 89]. Moreover, a cytotoxic indirect effect has been reported on Tregs and MDSCs, thereby enhancing the antitumor response [8, 90–92]. Therefore, we could suggest a synergic and bidirectional interaction between the immune system and NAC, where the anti-tumor effect of NAC is partially mediated by an immune effect which in turn is potentiated by the action of the treatment.

NAC affects the TIME, for example Demaria, et. al., demonstrated that Paclitaxel-based treatment increases the frequency of TILs in BC patients who responded to treatment [34]. TILs are a prognostic biomarker for clinical response [93, 94]. It is important to highlight that these studies focus on morphological evaluation of lymphocytes, without considering the expression of proteins and other different cell populations that may be involved in antitumor immunity. The prognostic value of CD8+ T cells in BC patients has been confirmed in other studies [95, 96].

In BC, Mφ, DCs, NKs, T and B cell infiltration has been reported. In addition, the expression of PD1, PD-L1, CTLA4 exhaustion markers and chemokines such as CXCL13 has been evaluated [94, 97, 98]. However, the role of these cellular and molecular markers in disease prognosis has not been fully elucidated. Most studies included in this review evaluated TILs, reported no significant differences, however, in the meta-analysis we found a significant decrease in TILs after NAC, which contradicts the hypothesis that NAC increases immune infiltration. However, because of the method used to measure TILs, it may include T and B cells, and even NKs that share the same cell morphology, let alone all their possible regulatory or effector phenotypes.

According to Pelekanou, et. al., the decrease in TILs in response to NAC could be explained by two hypotheses [24]. Firstly, that there is a cytotoxic effect of NAC on this population or, secondly that as tumor progression ceases, the target of the infiltrate is diminished. In contrast with the hypothesis that NAC triggers the immune infiltration by inducing DAMPs, an idea supported by Li, et. al., [25] and by Greco-Hoffman, et. al., [36]. It is important to analyze whether the antitumor immune response depends on the antigenic concentration in the TIME, since, when tumor cells decrease by direct cytotoxicity of NAC, indirectly also decreases the DAMPs and therefore the tumor infiltration capacity.

Expression of ICP such as: CTLA4, PD1, LAG3, and TIM3 regulates prolonged activation of T-cells by maintaining peripheral tolerance and preventing autoimmunity [99]. Tumors can exploit these control pathways to evade the immune response, as a matter of fact, PD-L1 expression correlates with tumor size, grade, metastatic spread, as well as with reduced levels of CD8+ T cells [100, 101]. Moreover, CTLA4 expression is higher in BC than in normal breast tissue and increased expression levels are associated with node metastasis and globally with higher tumor stage [99].

The articles included in this review do not uniformly report the trend of change of these ICP. CTLA4 decreases after NAC, however, this marker was evaluated only by one included article [37]. PD1 tends to increase in CD8+ T cells [60, 68], while decreases in CD4+ T cells [60]. If clinical response is considered, PD1 decreases only in those who reach pCR [60]. In most studies PD-L1 expression was either stable or it decreased after NAC, however, the only paper reporting an increase does so only in stromal cells [28]. LAG3 expression in CD8+ T cells increases after NAC in patients with poor clinical response [68] and generally decreases at the stromal level [38]. Considering the evaluation of different ICP, some studies suggest the clinical utility of jointly blocking PD1 and LAG3 [68, 102, 103]. Conflicting results of in-situ ICP changes probably arise from i) the heterogeneity of BC in terms of molecular profiles and oncogenic mechanisms among different types or subtypes of breast cancer [28], ii) cut-off points to establish positivity for certain markers, iii) the use of different anti PD-L1 antibodies

[71], and iv) differences in the activation status of T cells and their cytokine production in response to NAC [23].

Some articles included evaluated cellular and molecular marker changes in early response to chemotherapy (single cycle of NAC). Lee, et. al., and Hee-Park, et. al., observed an increase in intratumoral and stromal TILs levels after one cycle of NAC in patients who presented a favorable response [26, 28]. However, Hee-Park, et. al., reported a decrease in TILs at the end of the NAC regimen [26]. This behavior would suggest, as pointed before that, at first, NAC induces MCI and increases tumor infiltration, but as the tumor decreases in size, danger signals are reduced and the antitumor immune response resolves.

The expression of PD1 and PD-L1 and their association with the antitumor response present contradictory results. Graeser, et. al., reported a decrease of PD-L1 in early stages of therapy [62], which could be related to the elimination of tumor cells, and an increase of PD1 expression in immune cells that would suggest an activation of T cells [104, 105], however, Lee, et. al., found an increase in PD-L1 expression that correlated with a worse prognosis [28].

Regarding the innate immune system, the population that has been most studied in the context of tumor infiltration is TAMs (tumor associated Mφ). However, it has not been possible to reach a consensus on the markers that should be used to define this population.. No definitive trend in the change of global macrophage infiltration, neither M1 nor M2, in response to NAC was observed in the included articles. New studies have proposed novel markers such as TREM2 and FOLR2, which have shown a significant association with clinical response in patients and should be included in new studies in this field [83].

Articles measuring TINs, DCs and NK cells were also included.. Similarly, TINs, which are attracted to the tumor site by TAMs, have also been associated with poor DFS and OS [106].

NKs are the quintessential antitumor cells, and their high cytotoxic capacity explains their positive association with clinical factors [107, 108]. It was not possible to perform a meta-analysis between the two articles that reported percentages of NKs before and after NAC, and no trend could be found either, since one of them reported an increase in stroma in response to NAC [43], but in the other study no changes were found [39].

As for DCs,. three papers included measurement of these cells and, although a meta-analysis could not be performed, all three studies reported an overall increase in DCs in post-NAC samples [29, 43, 44]. Additionally, Hornychova, et. al., included CD83 and S100, DC maturation markers, suggesting that these cells are presenting antigen appropriately and report an increased expression of these in the stroma in response to NAC [43].

Components of the adaptive immune system such as CD4+ and CD8+ T cells, which recognize antigens and subsequently induce an effective immune response against cancer, can be found infiltrating the tumor. The presence of Tregs, which decrease the antitumor immune response, has been reported. In our study it was possible to perform meta-analyses for various subsets of T cells. Unfortunately, none of them, except for FoxP3+ cells, showed no significant differences between pre- and post-NAC (Figs 5–9). The meta-analysis done for Tregs showed a decrease in response to NAC, which can be explained due to their increased sensitivity to this type of NAC [109, 110]. Likewise, it has been proposed that decrease in Tregs after NAC can be used as a predictive marker for pCR [58, 65, 111, 112].

It is important to highlight that Tregs are heterogeneous cells, so identifying this population with Foxp3 alone may be insufficient [113–115]. This could be evidenced in Zhang Lin's study, in which both Foxp3 and CD25 were used as Tregs markers, and they found that double positive Tregs correlated with better OS, while the Foxp3+ alone subset did not, even though in both cases they were significantly reduced after NAC [71]. These results emphasized the importance of using more than one marker to identify subsets of Tregs.

With the meta-analyses performed a significant decrease in TILs and FoxP3 expression after NAC could be established. However, when dividing T cells into CD8+ and CD4+, no significant changes post-NAC could be established. Most studies evaluated the relationship between the presence of immune infiltrate components and clinical response, however, very few correlated the change in the infiltrate in response to NAC and the patient's clinical outcome, so conclusions related to prognosis could not be established.

The presence of CD8+ T cells alone is not sufficient to generate an effective antitumor response, because if high numbers of regulatory cells are present, they may suppress the cyto-toxic-mediated immune response and allow tumor growth. Therefore, studies propose to have indicators that consider both populations and measure the CD8/FoxP3 ratio [28, 66, 67]. The articles that report this indicator show that an increase in this ratio is related to a better clinical prognosis [65–67]. This ratio may be of great utility in determining subgroups of patients who may be suitable for other types of immunotherapies such as anti-checkpoint antibodies [60].

In the present study only the effect of NAC was evaluated, so all those articles dealing with other types of treatments were discarded. However, Waks, et. al., and Ladoire, et. al., had a cohort of patients with NAC and another with NAC + Bevacizumab and Trastuzumab and reported high concordance between the two cohorts at the level of FoxP3 decrease and CD8 + maintenance, which correlated with clinical outcome [29, 65].

Cytokines secreted by the cells that compose the TIME modulate the innate and adaptive immune response, however their role in the control or progression of cancer and in the sensitivity to NAC is not clearly established [116], partly because the dynamics of communication between tumor cells and immune cells through cytokines and chemokines is modified during cancer initiation, progression, and therapeutic intervention [117]. Furthermore, the complex cytokine system makes it difficult to attribute changes in their production and function as a cause or consequence of the neoplastic process or as an immune response against cancer development [117].

Only three included articles evaluated the presence of *in-situ* cytokines, and none evaluated chemokines although these have been related to immune cell migration and infiltration, as well as to angiogenesis and tumor stemness [117, 118]. In most cases, there were no significant changes in cytokine levels in response to treatment, with the exception of a decrease in IL-4 post-NAC and a trend of downregulation in IL-2 [37].

Joint decrease in IL-2 and IL-4 could be inducing a decrease in certain Tregs function after NAC [119], which would explain the stromal and tumor decrease in FoxP3 reported here, as well as the significant association between post-NAC expression of IL-10 and IL-17 and failure to obtain pCR [37].

Finally, we evidenced a high heterogeneity in the study and evaluation of the immune infiltrate in BC tumors. Classification of the type of tumor, the NAC schedule prescribed, and the differential measurement of TILs between the tumor and the surrounding stroma represent the variables with the greatest heterogeneity among studies (S3 Table), which generates great difficulty in performing meta-analyses.

Further study of the effect of NAC on the infiltrate will be important for the introduction of different immunotherapy strategies in the context of NAC, as well as to understand the mechanism and biological significance of such immunological changes.

## Supporting information

**S1 Checklist. PRISMA 2020 checklist.**
(DOCX)

**S1 Fig. Risk of bias summary by article.**
(TIF)

**S1 Appendix. Search strategy.**
(PDF)

**S2 Appendix. Legend of S1 Fig.**
(PDF)

**S1 Table. Included articles and parameters evaluated.**
(PDF)

**S2 Table. Excluded articles.**
(PDF)

**S3 Table. Heterogeneity.**
(PDF)

**S4 Table. TILs trends by article.**
(PDF)

**S5 Table. Cells trends by article.**
(PDF)

**S6 Table. Cytokines and immune checkpoints trends by article.**
(PDF)

## Author Contributions

**Conceptualization:** Manuela Llano-León, Laura Camila Martínez-Enriquez, Oscar Mauricio Rodríguez-Bohórquez, Esteban Alejandro Velandia-Vargas, Nicolás Lalinde-Ruíz, María Alejandra Villota-Álava, Ivon Johanna Rodríguez-Rodríguez, María del Pilar Montilla-Velásquez.

**Data curation:** Manuela Llano-León, Laura Camila Martínez-Enriquez, Oscar Mauricio Rodríguez-Bohórquez, Esteban Alejandro Velandia-Vargas, Nicolás Lalinde-Ruíz, María Alejandra Villota-Álava, María del Pilar Montilla-Velásquez.

**Formal analysis:** Manuela Llano-León, Laura Camila Martínez-Enriquez, Oscar Mauricio Rodríguez-Bohórquez, Esteban Alejandro Velandia-Vargas, Nicolás Lalinde-Ruíz, María Alejandra Villota-Álava, María del Pilar Montilla-Velásquez.

**Investigation:** Laura Camila Martínez-Enriquez.

**Methodology:** Manuela Llano-León, Oscar Mauricio Rodríguez-Bohórquez, Esteban Alejandro Velandia-Vargas, Nicolás Lalinde-Ruíz, María Alejandra Villota-Álava, Ivon Johanna Rodríguez-Rodríguez, María del Pilar Montilla-Velásquez.

**Supervision:** Carlos Alberto Parra-López.

**Writing – original draft:** Manuela Llano-León, Laura Camila Martínez-Enriquez, Oscar Mauricio Rodríguez-Bohórquez, Esteban Alejandro Velandia-Vargas, Nicolás Lalinde-Ruíz, María Alejandra Villota-Álava, María del Pilar Montilla-Velásquez.

**Writing – review & editing:** Manuela Llano-León, Laura Camila Martínez-Enriquez, Oscar Mauricio Rodríguez-Bohórquez, Esteban Alejandro Velandia-Vargas, Nicolás Lalinde-Ruíz, María Alejandra Villota-Álava, Ivon Johanna Rodríguez-Rodríguez, María del Pilar Montilla-Velásquez, Carlos Alberto Parra-López.

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
