## [Decision Letter · Decision Letter 0]

19 Dec 2022

PONE-D-22-27675Effect of neoadjuvant chemotherapy on tumor immune infiltration in breast cancer patients: systematic review and meta-analysis.PLOS ONE

Dear Dr. Parra-Lopez,

Thank you for submitting your manuscript to PLOS ONE. After careful consideration, we feel that it has merit but does not fully meet PLOS ONE’s publication criteria as it currently stands. Therefore, we invite you to submit a revised version of the manuscript that addresses the points raised during the review process.

We look forward to receiving your revised manuscript.

Kind regards,

Sumit Kumar Hira, Ph.D.

Academic Editor

PLOS ONE

Journal Requirements:

Reviewers' comments:

Reviewer's Responses to Questions

**Comments to the Author**

1. Is the manuscript technically sound, and do the data support the conclusions?

Reviewer #1: Yes

Reviewer #2: Yes

2. Has the statistical analysis been performed appropriately and rigorously? 

Reviewer #1: Yes

Reviewer #2: Yes

3. Have the authors made all data underlying the findings in their manuscript fully available?

Reviewer #1: Yes

Reviewer #2: Yes

4. Is the manuscript presented in an intelligible fashion and written in standard English?

Reviewer #1: Yes

Reviewer #2: Yes

5. Review Comments to the Author

Reviewer #1: In this manuscript, Llano-León et al. performed a systematic review and meta-analysis to evaluate the effect of neoadjuvant chemotherapy (NAC) in the immune infiltration of breast cancer tumors. Most of the studies evaluated in the manuscript reported no significant change in TILs after NAC. However, in the meta-analysis, the authors reported a significant decrease in TILs and FOXP3 after NAC. The authors reported a high heterogeneity in the study and evaluation of the immune infiltrate in breast cancer tumors. The authors’ analysis reveals the need for further study of the effect of NAC on the infiltrate which will be important for the introduction of different immunotherapy strategies.

The manuscript is carefully written, well organized, and does not contain any noticeable typographical error or grammatical mistake. References and abbreviations have been used appropriately. This manuscript should be of interest to a broad audience, in particular to the researchers and clinicians working on cancer therapy. However, I would like to mention the following issues that needs to be addressed.

1. Please provide a detailed description of the “Search strategy” so that it is understandable to a broader audience.

2.In Figure 2 legend, a), b), c), ….m) are not specified. I assume those are same as that of Figure3, please clarify.

3. Figure 3 shows the same results as that of Figure 2, only those are represented article-wise, without any substantial new information. Therefore, this figure 3 can go as a Supplementary figure.

4. In table-2, instead of presenting which article shows what TIL trend, it would be easier to understand if the authors present which TIL trend (increase, decrease, no change) appears in how many studies with a simple bar graph.

5. Too many information are spread across Table-3 which makes it difficult to comprehend. Again, instead of showing cell type trends in each article separately, it would be easier to follow if the authors show which cell type trend appears in how many studies with a graph if possible.

6. Since the present study has been limited only to neoadjuvant chemotherapy, very few studies qualified for meta-analysis. It might provide a broader picture if other neoadjuvant therapies (immunotherapy, hormone therapy etc.) are included and the findings may be clinically more significant.

Reviewer #2: PONE-D-22-27675

"Effect of neoadjuvant chemotherapy on tumor immune infiltration in breast cancer patients: systematic review and meta-analysis."

Tumor microenvironments are complex structures comprised of various interconnected layers that are difficult to define and study. The tumor immune microenvironment (TIME), in particular, refers to the presence of leukocytes, as well as their products, surface markers, and gene expression profiles within or around a tumor. The variability in the presence, location, and functional organization of infiltrating immune cells suggests that different populations may play distinct roles in the control or promotion of tumor growth. The study of these characteristics has led to the identification of prognostic markers for clinical response to treatment, resulting in the development of different approaches to classify neoplasms. In recent years, the validation of models such as Immunoscore has allowed for a more refined evaluation of patients' prognoses, particularly in colorectal cancer. In breast cancer (BC), the importance of immune infiltration as a prognostic factor for clinical response to neoadjuvant chemotherapy (NAC) has also been evaluated. Chemotherapy has traditionally been seen as an immunosuppressive treatment, but evidence suggests that it can actually activate the immune system by inducing immunogenic cell death in tumor cells, leading to the release of DAMPs.

This review aims to determine whether there are changes in the cellular and molecular components of the immune infiltrate in BC tumours in response to NAC. A systematic search of the literature was performed, and studies comparing immune molecules or cells before and after treatment were included. The results showed that NAC leads to an increase in the expression of immune molecules such as HLA-DR, PD-L1, and CD8, as well as a decrease in the expression of immune inhibitory molecules such as PD-1, CTLA-4, and Tim-3. There was also an increase in the infiltration of immune cells such as T lymphocytes, dendritic cells, and macrophages in response to NAC. However, these findings are not universal and are not present in all studies. Some papers showed there is no correlation between immune cells and NAC, while others showed the opposite result. Due to the heterogeneity among the studies, there is no conclusive evidence that suggests NAC can lead to a more favourable.

Liano-Leon et al addressed an important question, though there is no conclusion. It is still important to consider this topic for future study designs. However, although they thoroughly checked most of the available literature, I would not recommend accepting the current form of the manuscript. If the work is to be accepted for publication, I suggest addressing the following points:

Major Comments:

1. The English of the manuscript is generally acceptable, but the structure of the manuscript could be improved to make it more reader-friendly, particularly in the Results and Discussion sections. Those can be written in a more concise way. There is a lot of redundancy in the Discussion and Results sections. The authors could consider cutting and pasting some text from the Discussion section to the Results section in order to improve the overall structure and clarity of the paper. It may be helpful to include some background information on the cells or molecules being studied in the Results section, as this can provide context and make the results more meaningful for readers who are not familiar with them. Additionally, it would be beneficial to include an overall conclusion or implication of the results at the end of the Results section. This could highlight any trends observed or speculations that can be made based on the data.

2. Line 150-157: The Table 1 can be found in the supplementary section. A Venn diagram can provide a visual representation of the table's content, which may be more appealing to readers than a lengthy table.

3. Line 191- 216: Table 2, which describs all the TILs’ results can be moved to the supplementary section. Forest plots or funnel plots may be easy more easily understood by the readers.

4. Similarly, Table 3 can be moved to the supplementary section and forest plots or funnel plots may be more easily understood by readers.

5. The authors concluded that there was a significant decrease in TILs (tumor infiltrating lymphocytes) and Foxp3 expression after NAC. I am wondering if TILs and Foxp3 can be used as prognostic markers for DFS after NAC. Is it possible to develop a logistic regression equation using these variables?

Minor Comments:

1. Fig 2 is enough for main figure. Fig 3 can be found in the supplementary section. The description of a-m should be in Fig 2 legend.

2. Line 233: What is NOtcWEh1? Is it a typo of Notch1? I did not find it in the reference (49).

3. Line 110-IFN-y, Line 50: IFNg. Are authors mentioning IFN-gamma (IFNg) at Line 110?

6. PLOS authors have the option to publish the peer review history of their article (what does this mean?). If published, this will include your full peer review and any attached files.

Reviewer #1: **Yes: **SAMIT CHATTERJEE

Reviewer #2: No

---

## [Author Response · Author response to Decision Letter 0]

23 Mar 2023

Bogota, February 28, 2023

Review:

PONE-D-22-27675

"Effect of neoadjuvant chemotherapy on tumor immune infiltration in breast cancer patients: systematic review and meta-analysis."

PLOS ONE

Dear academic editor and reviewer(s),

In accordance with your observations and suggestions, we allow ourselves to respond to the following points through the next table:

Reviewer 1 comments:

1. Please provide a detailed description of the “Search strategy” so that it is understandable to a broader audience.

We have described in more detail the Search strategy in lines 81-87 in the non-tracked version. 

2. In Figure 2 legend, a), b), c), ….m) are not specified. I assume those are same as that of Figure3, please clarify. 

We describe the a to m variables in the figure 2. 

3. Figure 3 shows the same results as that of Figure 2, only those are represented article-wise, without any substantial new information. Therefore, this figure 3 can go as a Supplementary figure.

We change Figure 3 to Figure S1.

4. In table-2, instead of presenting which article shows what TIL trend, it would be easier to understand if the authors present which TIL trend (increase, decrease, no change) appears in how many studies with a simple bar graph.

Table 2 were replaced with table 1. This table includes bar graphs presenting TILs and cells trends. 

5. Too many information are spread across Table-3 which makes it difficult to comprehend. Again, instead of showing cell type trends in each article separately, it would be easier to follow if the authors show which cell type trend appears in how many studies with a graph if possible.

Table 3 were replaced with table 1. This table includes bar graphs presenting TILs and cells trends.

6. Since the present study has been limited only to neoadjuvant chemotherapy, very few studies qualified for meta-analysis. It might provide a broader picture if other neoadjuvant therapies (immunotherapy, hormone therapy etc.) are included and the findings may be clinically more significant.

The aim of this review was to evaluate the effect of neoadjuvant chemotherapy on the tumor immune infiltrate; therefore we defined the combination with other therapies as an exclusion criteria. Including these therapies would imply carrying out another search strategy that would not meet our aim. However, we will consider it for future works, as the reviewer's comment seems very relevant to us.

Reviewer 2 comments:

Major comments:

1. The English of the manuscript is generally acceptable, but the structure of the manuscript could be improved to make it more reader-friendly, particularly in the Results and Discussion sections. Those can be written in a more concise way. There is a lot of redundancy in the Discussion and Results sections. The authors could consider cutting and pasting some text from the Discussion section to the Results section in order to improve the overall structure and clarity of the paper. It may be helpful to include some background information on the cells or molecules being studied in the Results section, as this can provide context and make the results more meaningful for readers who are not familiar with them. Additionally, it would be beneficial to include an overall conclusion or implication of the results at the end of the Results section. This could highlight any trends observed or speculations that can be made based on the data.

We move sections of the discussion to the results and made a brief description of the cells and markers evaluated. The new texts are found in the following lines:

• TILs: lines 195-196

• Macrophages: lines 244-246

• Neutrophils: lines 253-255

• Natural killer cells: lines 261-263

• Dendritic cells: lines 292-295

• B cells: lines 287-290

• T cells: lines 297-302

• CD4 T cells: lines 321-323

• CD8 T cells: lines 342-344

• FoxP3: lines 383-385

• Immune checkpoint and cytokines: lines 417-427; lines 460-472; lines 476-478.

Finally, we added a conclusion paragraph in lines 499-510 at the end of the result section. 

2. Line 150-157: The Table 1 can be found in the supplementary section. A Venn diagram can provide a visual representation of the table's content, which may be more appealing to readers than a lengthy table.

We replace Table 1 with a waffle plot where each box represents an article which measure a type of cell or a marker. This table was move to the supplementary information as Table S1. 

3. Line 191- 216: Table 2, which describs all the TILs’ results can be moved to the supplementary section. Forest plots or funnel plots may be easy more easily understood by the readers.

Table 2 were replaced with table 1. This table includes bar graphs presenting TILs and cells trends. 

4. Similarly, Table 3 can be moved to the supplementary section and forest plots or funnel plots may be more easily understood by readers.

Table 3 were replaced with table 1. This table includes bar graphs presenting TILs and cells trends.

5. The authors concluded that there was a significant decrease in TILs (tumor infiltrating lymphocytes) and Foxp3 expression after NAC. I am wondering if TILs and Foxp3 can be used as prognostic markers for DFS after NAC. Is it possible to develop a logistic regression equation using these variables? 

Quantitative data was extracted directly from the graphs through the Graph grabber extraction program, since this information was requested to the authors of each article without receiving any response. Therefore, we do not have the raw data paired with the clinical outcome of the patients and we cannot perform a meta-regression to define the prognostic value of these variables.

Minor comments:

1. Fig 2 is enough for main figure. Fig 3 can be found in the supplementary section. The description of a-m should be in Fig 2 legend.

We describe the a to m variables in the figure 2 and move Figure 3 to the supplementary information as Figure S1.

2. Line 233: What is NOtcWEh1? Is it a typo of Notch1? I did not find it in the reference (49).

NotcWEh1 was a typo, we corrected it to Notch1.

3. Line 110-IFN-y, Line 50: IFNg. Are authors mentioning IFN-gamma (IFNg) at Line.

We change all forms of IFN-gamma to IFNg. 

Thank you to the reviewers for their valuable feedback. We have carefully considered and incorporated your comments and look forward to hearing back from the editors regarding the final decision on our manuscript. 

Sincerely,

Carlos Alberto Parra López MD., Ph. D.

Full Professor School of Medicine

Microbiology Department

Universidad Nacional de Colombia

---

## [Decision Letter · Decision Letter 1]

13 Apr 2023

Effect of neoadjuvant chemotherapy on tumor immune infiltration in breast cancer patients: systematic review and meta-analysis.

PONE-D-22-27675R1

Dear Dr. Parra-López,

We’re pleased to inform you that your manuscript has been judged scientifically suitable for publication and will be formally accepted for publication once it meets all outstanding technical requirements.

Kind regards,

Sumit Kumar Hira, Ph.D.

Academic Editor

PLOS ONE

Additional Editor Comments (optional):

Reviewers' comments:

Reviewer's Responses to Questions

**Comments to the Author**

1. If the authors have adequately addressed your comments raised in a previous round of review and you feel that this manuscript is now acceptable for publication, you may indicate that here to bypass the “Comments to the Author” section, enter your conflict of interest statement in the “Confidential to Editor” section, and submit your "Accept" recommendation.

Reviewer #1: All comments have been addressed

Reviewer #2: All comments have been addressed

2. Is the manuscript technically sound, and do the data support the conclusions?

Reviewer #1: Yes

Reviewer #2: Yes

3. Has the statistical analysis been performed appropriately and rigorously? 

Reviewer #1: Yes

Reviewer #2: Yes

4. Have the authors made all data underlying the findings in their manuscript fully available?

Reviewer #1: Yes

Reviewer #2: Yes

5. Is the manuscript presented in an intelligible fashion and written in standard English?

Reviewer #1: Yes

Reviewer #2: Yes

6. Review Comments to the Author

Reviewer #1: The authors have addressed the queries and substantially improved the manuscript. Therefore, it may be accepted.

Reviewer #2: After carefully reviewing the revised manuscript, I am pleased to report that the authors have adequately addressed all of my concerns. The revisions have significantly improved the quality and clarity of the manuscript, and I believe that it is now suitable for publication in Plos One.

7. PLOS authors have the option to publish the peer review history of their article (what does this mean?). If published, this will include your full peer review and any attached files.

Reviewer #1: **Yes: **SAMIT CHATTERJEE

Reviewer #2: No

---

## [Editor Report · Acceptance letter]

18 Apr 2023

PONE-D-22-27675R1 

Effect of neoadjuvant chemotherapy on tumor immune infiltration in breast cancer patients: systematic review and meta-analysis. 

Dear Dr. Parra-López:

I'm pleased to inform you that your manuscript has been deemed suitable for publication in PLOS ONE. Congratulations! Your manuscript is now with our production department. 

Kind regards, 

on behalf of

Dr. Sumit Kumar Hira 

Academic Editor

PLOS ONE